# Bending Fracture of Different Zirconia-Based Bioceramics for Dental Applications: A Comparative Study

**DOI:** 10.3390/ma14226887

**Published:** 2021-11-15

**Authors:** Peter Fischer, Horia Mihail Barbu, Caroline Adela Ingrid Fischer, Mihaela Pantea, Florin Baciu, Diana Maria Vranceanu, Cosmin Mihai Cotrut, Tudor Claudiu Spinu

**Affiliations:** 1Oral Implantology Department, Titu Maiorescu University, 67A Gh. Petrascu Street, 040441 Bucharest, Romania; peter@fischer.ro (P.F.); horia.barbu@gmail.com (H.M.B.); 2Doctoral School, Faculty of Dental Medicine, Titu Maiorescu University, 67A Gh. Petrascu Street, 040441 Bucharest, Romania; caroline.fischer@ymail.com; 3Department of Fixed Prosthodontics and Occlusology, Faculty of Dental Medicine, Carol Davila University of Medicine and Pharmacy, 17-23 Plevnei Street, 010221 Bucharest, Romania; mihaela.pantea@umfcd.ro (M.P.); tudor.spinu@umfcd.ro (T.C.S.); 4Department of Strength of Materials, Faculty of Industrial Engineering and Robotics, University Politehnica of Bucharest, 313 Splaiul Independentei, 060042 Bucharest, Romania; florin.baciu@upb.ro; 5Department of Metallic Materials Science, Physical Metallurgy, Faculty of Materials Science and Engineering, University Politehnica of Bucharest, 313 Splaiul Independentei, 060042 Bucharest, Romania

**Keywords:** zirconia, bioceramic, bending test, dental applications

## Abstract

The fabrication of fixed dental prostheses using aesthetic materials has become routine in today’s dentistry. In the present study, three-unit full zirconia fixed prosthetic restorations obtained by computer-aided design/computer-aided manufacturing (CAD/CAM) technology were tested by bending trials. The prostheses were intended to replace the first mandibular left molar and were manufactured from four different types of zirconia bioceramics (KatanaTM Zirconia HTML and KatanaTM Zirconia STML/Kuraray Noritake Dental Inc.; NOVAZir^®^ Fusion float^®^ ml/NOVADENT/Dentaltechnik; and 3D PRO Zirconia/Bloomden Bioceramics). In total, sixteen samples were manufactured—four samples per zirconia material. Additionally, the morphology, grain size area distribution, and elemental composition were analyzed in parallelepiped samples made from the selected types of zirconia in three different areas, noted as the upper, middle, and lower areas. The scanning electron microscope (SEM) analysis highlighted that the grain size area varies with respect to the researched area and the type of material. Defects such as microcracks and pores were also noted to a smaller extent. In terms of grain size area, it was observed that most of the particles in all samples were under 0.5 μm^2^, while the chemical composition of the investigated materials did not vary significantly. The results obtained after performing the bending tests showed that a zirconia material with fewer structural defects and an increased percentage of grain size area under 0.5 µm^2^, ranging from ~44% in the upper area to ~74% in the lower area, exhibited enhanced mechanical behavior. Overall, the resulting values of all investigated parameters confirm that the tested materials are suitable for clinical use.

## 1. Introduction

One’s public image is becoming an important facet of contemporary society. Satisfying aesthetic needs complements medical treatment and, as a consequence, has become common practice in prosthetic dentistry [1,2,3,4], along with the fabrication of dental prostheses using aesthetic materials. On the other hand, the use of resistant, conventional materials in prosthetic dentistry, such as different metal alloys, can cause side effects, such as possible allergies and negative perceptions of patients regarding their aesthetic results [5,6]. Finding a material with both aesthetic qualities and mechanical toughness has always been a challenge [7]. Leucite-reinforced glass-ceramics or lithium disilicate glass-ceramics have been considered as standard in aesthetics for many years; however, their fragility only allows for the fabrication of single-unit restorations or short-span dental prostheses, comprising a maximum of three units—up to the second premolar as the distal abutment—as in the case of lithium disilicate [8,9]. Thus, developing materials strong enough to successfully replace the long-span metallic frameworks has become a challenge in modern scientific research. The new generations of zirconia and ceramic materials are now widely used in restorative dentistry due to their attractive appearance, superior mechanical properties, and biocompatibility [10]. The clinical and laboratory workflow in dentistry has dramatically improved as a result of the development of computerized systems (computer-aided design/computer-aided manufacturing; CAD/CAM) dedicated to the design and manufacture of diverse dental prostheses, as well as following the introduction of new generations of high-strength ceramic materials, zirconia-based ceramic standing out as one of the most representative [10,11,12,13].

Concerns regarding the use of aesthetic restorations in the posterior area of the dental arches have been related to the mechanical and optical properties of the available dental materials [14]. Veneering zirconia infrastructures with feldspathic ceramics have been associated with mechanical complications such as chipping and delamination (15–36% for veneered glass-ceramics compared to only 0–8% failures in the case of zirconia frameworks, over a period of 5 years) [15]. Because of the strong occlusal forces acting in the posterior areas, cohesive fractures of veneering ceramics occur more frequently than failures of zirconia infrastructures, which have demonstrated excellent fracture resistance [16]. Consequently, the use of monolithic zirconia restorations has become an attractive alternative due to improved aesthetic appearance, good mechanical behavior, a simplified technological process, and the advantage of a more conservative tooth preparation [14,17,18,19].

Monolithic zirconia crowns in the posterior area showed optimal marginal adaptation, cervical margin integrity, and color match. Very promising clinical results were also reported regarding the influence on periodontal tissues and the wear of antagonist teeth [20].

Analyzing the fracture resistance of 136944 zirconia restorations (single crowns and fixed partial dentures) in both anterior and posterior areas, Sulaiman et al. [21] observed that, for up to 7.5 years, layered restorations fractured more frequently than monolithic ones in both areas. Furthermore, monolithic crowns and fixed prostheses had the lowest fracture incidence (0.54% and 1.83%, respectively) compared to layered fixed prostheses (1.93%) and crowns (2.83%). Some encouraging results have also been reported regarding the use of monolithic zirconia for full-arch implant-supported fixed prostheses [22].

The main motivations for using the newly developed zirconia-based materials have been to improve the appearance of restorations, maintain their material strength so that they should withstand mechanical stress, and preserve their internal and marginal fit, chemical stability, and biocompatibility in the complex oral environment [14,23,24,25]. Manufacturers have also succeeded in improving the optical characteristics by introducing multilayer zirconia discs with increased translucency, which, to a certain extent, has led to a reduction in the flexural strength [26]; however, this material presents acceptable fracture load values compared to the maximum occlusal bite force [24]. Multilayer zirconia exhibits aesthetic and mechanical properties in-between conventional zirconia and lithium disilicate, being recommended for individual anterior teeth restorations and full-mouth prosthetic rehabilitations as well [27,28]. Moreover, the optical and mechanical properties of multilayer zirconia material are influenced by factors such as grain and pore size, configuration of the crystals, chemical composition, thickness, and surface treatment.

The main objectives of the present study were to investigate the morphology, chemical composition, and fracture resistance of certain newly developed multilayered zirconia materials with augmented optical and mechanical properties. These materials were examined by means of scanning electron microscopy and energy-dispersive spectroscopy. Three-unit fixed dental prosthesis samples were afterwards manufactured from these materials and subjected to bending tests. The null hypothesis was stated as follows: the new zirconia-based ceramic materials with progressively increased translucency exhibit results comparable with conventional zirconia specimens with constant flexural strength.

## 2. Materials and Methods

### 2.1. Sample Preparation

In the current study, four types of zirconia-based materials were investigated. Table 1 presents the details corresponding to all tested materials.

KATANA^TM^ Zirconia HTML (high-translucency multilayered) and STML (super-translucent multilayered) are produced by Kuraray Noritake Dental Inc. (Tokyo, Japan), providing, according to the manufacturer, a constant flexural strength of approximately 1100 MPa and 750 MPa, respectively. NOVAZir^®^ Fusion float^®^ ml (NOVADENT Dentaltechnik, Göttingen, Germany) is a multilayered zirconia disc with a flexural strength ranging from 800 MPa in the occlusal area up to 1100 MPa in the cervical area. 3D PRO Zirconia from Bloomden Bioceramics (Hunan, China) is also a multilayered disc, with a flexural strength varying from 650 to 1100 MPa.

### 2.2. Morphology and Elemental Composition

The morphology and elemental composition were examined by SEM equipped with an energy-dispersive spectrometer (EDS) (Phenom PRO X, Phenom World). These characteristics were studied on three different areas of the parallelepiped samples obtained from the zirconia discs by using CAD/CAM technology, to observe the grain modifications. Figure 1 presents macroscopic images of the parallelepiped samples before and after sintering, along with their dimensions and an illustration of the sections to be investigated by SEM and EDS on sintered samples.

### 2.3. Bending Test

Three-unit zirconia dental prostheses corresponding to the replacement of the first mandibular left molar were manufactured by computer-aided design/computer-aided manufacturing (CAD/CAM) technology and then subjected to bending tests. A Co-Cr dental alloy (NOVADENT/Dentaltechnik) abutment model, featuring the absence of the first mandibular left molar, was first obtained by CAD/CAM technology and, based on it, the zirconia samples were designed. Using Trios Design Studio software (3Shape), an STL file containing the three-unit prosthesis design was generated and imported into the CAM software. To avoid any force deflection that may be generated by the cuspal inclines, the anatomic occlusal surface was leveled into a horizontal plateau, using Meshmixer (Autodesk Inc., Mill Valley, CA, USA). This design can mimic the flatter occlusal surfaces, with reduced cusp inclinations, that are specific to prosthetic restorations recommended for clinical cases of old partial edentulism, for patients with bruxism or with a pattern of horizontal masticatory movements.

In the nesting process, the designed prosthesis models were positioned in the entire-disc thickness, addressing all material layers (Figure 2).

The experimental samples were obtained with a CORITEC 350i (Imes-Icore, Eiterfeld, Germany) milling machine and then sintered according to each manufacturer technical data sheet in an MIHM-VOGT HT sintering furnace (MIHM-VOGT GmbH & Co, Stutensee, Germany).

In this study, four types of zirconia-based materials were tested (Katana^TM^ Zirconia HTML and Katana^TM^ Zirconia STML/Kuraray Noritake Dental Inc; NOVAZir^®^ Fusion float^®^ ml/NOVADENT/Dentaltechnik; and 3D PRO Zirconia/Bloomden Bioceramics) (Figure 3). Respectively, four samples of a three-unit fixed dental prosthesis were fabricated from each material and investigated by a bending test. Therefore, 16 samples were finally manufactured and tested.

The bending resistance of the zirconia specimens was studied by applying a three-point bending test using an Instron 8872 Universal Testing Machine (Instron Inc., Norwood, MS, USA) equipped with a load cell of 25 kN. The testing speed was set to 1mm/min. A custom-made support was manufactured, and the load was applied on the pontic of the zirconia dental prostheses, using a plate for the purpose of reproducing the occlusal force (Figure 4).

The force was applied in the middle of the pontic and the test was carried out until a fracture occurred. Images and slow-motion videos were recorded for each tested sample.

## 3. Results

### 3.1. Morphology

The SEM images of the investigated specimens at different magnifications are presented in Figure 5. The SEM analysis revealed the presence of defects, microcracks, and pores (SEM images at a smaller magnification), as well as the fact that the grain size area and its distribution vary from one manufacturer to another. However, the B zirconia samples stand out as having only some minor defects, irrespective of the examined area (upper, middle, or lower), compared to the other investigated materials, whose specimens present noticeable defects.

All samples present both large and small grains, differently distributed throughout the material. The main findings with respect to the grain dimension are presented below:-The K-H specimens present both large and small grains, randomly distributed;-The B specimens present larger grains in the upper region, the grain size decreasing gradually to the lower region;-The N specimens have a multilayered structure regardless of the examined area, which consists of one layer with smaller grains, followed by a layer with larger ones;-Compared to the other investigated materials, the K-S specimens present the largest grains.

Because the morphology of zirconia consists of grains with irregular shapes, the grain size area (µm^2^) was measured with Image J software by using the SEM images, and the results are presented in Table 2 and Figure 6. To obtain more accurate information, all measurements were performed on areas of 27 × 27 µm^2^, in all three of the different regions of each sample, as indicated in Figure 1. The grain size area distribution was attained by selecting different intervals, given the grain dimension variation. Most of the grains were found to be equal to or under 1 µm^2^.

Since the three investigated areas of the K-H and N zirconia samples did not exhibit significant differences in terms of grain size (Figure 6), only the values obtained for just one section are presented. On the other hand, given the notable grain size differences between the analyzed sections of the B and K-S samples, all the related results have been illustrated in Table 2.

The grain size distribution analysis (Figure 6) indicates that the K-H and N zirconia samples are similar, with approximately 96% of the examined sections consisting of particles with a maximum size of 1 µm^2^ [29].

Nevertheless, while particles are randomly distributed in the K-H samples, larger grains being surrounded by smaller ones, in the N samples the particles are distributed incrementally (one layer of small-sized grains is followed by one consisting of larger-sized particles). This layered distribution was observed in all three investigated sections of sample N.

It was noticed that the B and K-S zirconia samples exhibited a dissimilar particle distribution compared to the other two specimens. The B zirconia samples presented different percentages of particles whose grain size is below 0.5 µm^2^, ranging from ~44% in the upper region (occlusal) to a maximum of ~74% in the bottom region (cervical), while the percentage of larger particles (whose area is higher than 0.5 µm^2^) decreases when passing from the upper to the lower section. When passing to the lower area, most of the particles were below 3 µm^2^.

On the other hand, regardless of the investigated area, the K-S zirconia samples presented the lowest percentage of particles whose area was below 0.5 µm^2^ (a min. of ~56% in the middle area and a max. of ~60% in the upper area).

Overall, the distribution of the grain size follows the same trend when passing from the upper to the lower area for all four types of zirconia included in our study. Moreover, it can be noted that by advancing from the upper region to the bottom one, the percentage of particles had the same trend (Figure 6).

### 3.2. Chemical Composition

The elemental distribution and chemical composition obtained on the basis of the EDS analysis performed on three different (upper, middle, and lower) areas of zirconia parallelepiped samples are indicated in Figure 7 and Figure 8.

The research revealed that elements are uniformly distributed in all investigated materials (Figure 7). It was also noted that all the zirconia samples presented similar values of the Zr element, regardless of the investigated area, while small variations were detected for yttrium (Y) and hafnium (Hf) (Figure 8). Therefore, it can be concluded that the examined specimens do not present significant differences regarding chemical composition.

### 3.3. Bending Test

Each zirconia sample corresponding to the three-unit dental prostheses was placed in the custom-made support and the steel plate was positioned on the central area of its surface, as shown in Figure 4.

The bending test was carried out by applying a 1 mm/min crosshead speed until fracture occurred. The load–displacement curves are presented in Figure 9a. Figure 9b indicates the force at the average failure values obtained for each zirconia material used to manufacture the three-unit dental bridges. For each zirconia type, four measurements were carried out.

After the bending tests, it was observed that all zirconia samples exhibited brittle failure in which the fragments perfectly fitted each other along the fracture line. Immediately after the initial fracture, total failure has occurred. In Figure 10 are presented the representative fractures lines observed for the zirconia 3-unit dental protheses. These findings are in good agreement with the study conducted by Lopez-Suarez et al [30]. 

As a general observation, the zirconia materials act differently under the same experimental conditions. Thus, the highest force at failure values were recorded for the B zirconia prostheses (2.00 ± 0.14 kN), followed by K-H and N samples, while the lowest values were obtained for the K-S specimens (Table 3). 

With respect to the mechanical tests performed, it can be assumed that the bending strength of the three-unit zirconia dental prostheses is influenced by the following critical factors: the size, shape, and distribution of particles in addition to the presence of material defects. The chemical composition of the tested materials does not appear to influence their mechanical properties as there are no significant variations in the concentration of elements, regardless of the investigated area or material.

B zirconia samples exhibited the highest force at failure values and thus the best mechanical behavior of all the investigated specimens. By corroborating the results of the bending tests with the grain size distribution presented in Table 2, the following observations were made:-The best results were obtained for sample B as its investigated upper and middle areas consist of smaller (≤1 µm^2^) and larger particles (between 1 and 5 µm^2^), while the lower areas are comprised of a very large number of grains whose maximum size is 2 µm^2^;-K-S zirconia samples presented the largest grain size, irrespective of the investigated areas, concomitantly with the lowest force at failure values.

The videos recorded (please see Appendix A) during the bending tests enabled the analysis of the fracture mechanism for each of the zirconia samples, starting with the initial crack and ending with the complete fracture. The 2D frames that were selected to indicate the fracture patterns for the tested zirconia types are presented in Figure 11.

Figure 11c, corresponding to the B samples, shows a fracture line from the occlusal area to the middle third of the crown towards the interproximal area, not reaching the cervical area. The other samples (K-H, N, and K-S) tend to exhibit a fracture line from the occlusal surface to the cervical area of the crown. A possible explanation could be that the mechanical strength decreases towards the occlusal surface in the multilayered restorations. Although thinner, the material in the cervical area is harder (1100 MPa), which leads to the presumption that hardness plays a more important role than thickness, combined with a decreased mechanical strength (down to 650 MPa) found in the middle and occlusal third of the crown. Nevertheless, this hypothesis needs to be confirmed by tests performed on a larger number of samples.

## 4. Discussion

During the previous decades, zirconia ceramics have been proven to be a reliable material for various types of prosthetic restorations. Additionally, due to their excellent bio-mechanical and optical properties, they have become one of the preferred choices for clinicians [31].

Zirconia ceramics were initially used as a substitute for metal alloys to obtain frameworks that were subsequently veneered with feldspathic ceramics. However, these bilayer structures were vulnerable to chipping and the delamination of veneered ceramics, thus affecting their long-term survival [5]. Due to these aspects, practitioners and producers were determined to turn to monolithic anatomic restorations instead, which had enhanced mechanical strength and toughness as well as greater simplicity in their fabrication [23,32].

Various yttria-stabilized zirconia ceramics are currently used as restorative materials in dentistry, their types varying depending on yttrium and additive contents as well as on the sintering conditions. Three mol% yttria-stabilized tetragonal zirconia (3Y-TZP) benefits from transformation toughening and has demonstrated excellent mechanical properties, such as high strength (prevention of crack initiation) and toughness (damage tolerance upon the occurrence of cracks) [33]. However, 3Y-TZP ceramics are susceptible to hydrothermal aging, which includes spontaneous phase transformation from tetragonal (T) to monoclinic (M) zirconia at moderate temperatures in a humid environment, such as the oral cavity or human body conditions. The opaqueness of 3Y-TZP ceramics is also a disadvantage [34]. In order to obtain more translucent 3Y-TZP ceramics, it was recommended to limit the alumina content [33], even if traces of alumina are acknowledged to improve the durability and stability of zirconia exposed to high temperatures and humid environments [35]; subsequently, partially stabilized zirconia (PSZ) containing a higher content of yttria (4 or 5 mol%) has been recently used to additionally increase the translucency of zirconia ceramics [33].

The above-mentioned positive aspects related to the mechanical behavior of monolithic zirconia rely on the specific characteristics of the material, which has a polymorphic crystalline structure that varies with temperature. In contrast to other ceramic materials, zirconia can pass from one crystalline phase to another during the thermal process [13]. The crystallographic phases of zirconia are the following: the monoclinic phase (M) occurs at room temperature up to 1170 °C and is characterized by a reduced mechanical resistance, influencing the cohesion of ceramic particles; the tetragonal phase (T) occurs from 1170 °C to 2370 °C, during which the material exhibits high mechanical resistance; and the cubic phase (C) occurs at temperatures above 2370 °C, characterized by a moderate mechanical resistance [36].

The transformation toughening, which is specific to zirconia, has proved useful, especially when the initial cracks occur. The stress contraction produced by the crack dispersion in the matrix represents a triggering factor, causing the particles that were uniformly dispersed during the T phase to pass into the M phase (martensitic transformation) [37]. This phenomenon that occurs around the crack is accompanied by an increment of approximately 4.4–4.5% of the volume of crystalline grains, which can cause the crack to constrict in addition to material hardening [37,38]. If oxides such as yttria (Y_2_O_3_), magnesia (MgO), ceria (CeO_2_), or calcia (CaO) are added, stabilized tetragonal zirconia polycrystals are obtained [13,39,40]. The stabilizing oxides allow for the preservation of the metastable tetragonal structure at room temperature. Grinding or sandblasting can induce a transformation in the surface from the T phase to the M phase, accompanied by a substantial increase in volume (~4.5%) which, in turn, generates surface compressive stresses, thereby closing the initial crack and limiting further propagation [35]. However, zirconia-based ceramics require special attention while performing adjustments when grinded; thus, it is recommended to apply light sawing movements in one direction, without using excessive force, and observing the anatomy of the area, using fine grit diamonds, under continuous water cooling. Moreover, it has been found that there is a significant correlation between zirconia mechanical properties (including phase stability) and its the crystalline grain size.

High flexural strength was reported for glass-infiltrated zirconia-toughened alumina (ZTA), characterized by a microstructure with large alumina grains (6 µm long, 2 µm wide), small zirconia grains (<1 µm in diameter), and few faceted zirconia grains (2 µm); trans-granular crack patterns and intra-granular patterns were observed for ZrO_2_ and Al_2_O_3_, respectively [38]. The advancing crack causes the T → M transformation while the increase in volume gives rise to microcracks in the alumina matrix surrounding the transformed particle; therefore, toughness is enhanced by microcracking. It can be concluded, therefore, that a larger grain size leads to an unstable material and spontaneous transformation into the M phase, whereas a smaller grain size inhibits the transformation, reducing the fracture toughness [38].

As cited in the dental literature [29], 3Y-TZP is less stable and more susceptible to a spontaneous transformation from the T phase to the M phase if the grain size is above 1 μm. On the other hand, the transformation toughening is not possible if the grain size is below 0.2 μm, which leads to reduced fracture toughness [29,35]. The mechanical properties of 3Y-TZP depend on the grain size, which is influenced by the sintering temperature: larger grain sizes occur when a higher temperature is maintained for a longer time during the sintering process.

Other studies [41,42,43] point out that a large particle size leads to a poor bending resistance, while a small particle size results in a higher bending force. Our results show that the K-S samples for which the lowest force at failure values were obtained also present significantly lower percentages of particles whose size is below 1 µm^2^, which corresponds to the manufacturer’s specifications regarding material strength for all samples. Furthermore, a small grain size has a positive effect on the polish potential of the final prosthesis [12].

Along with other studies which highlighted the fact that the strength of three-unit zirconia bridges is influenced by the design and connector geometry [44], the aim of the present study was to evaluate the force at failure by applying bending tests on three-unit bridges manufactured from different types of zirconia by CAD/CAM technology. On the other hand, the mechanical tests performed in the present study were carried out on actual fixed dental prostheses rather than on rectangular samples [39,45,46]; therefore, the obtained results offer valuable practical information and a specific perspective on a more realistic overview.

All the investigated samples presented defects, and the stress intensity factor was affected by the defects´ shape, depth, and width. Microcracks and defects that inherently grow during the thermal and mechanical processes can significantly influence the measurement of fracture resistance [47,48,49]. As cited in the literature, the strength and fracture toughness of the sintered zirconia are grain-size-dependent and can also be influenced by dopant concentration [49], while translucency is related to grain size, impurities, pores, and defects [31]. Nevertheless, all zirconia materials investigated in the present study exhibited fracture strength values that are higher than the clinically accepted ones.

Considering the recommendation of these materials for fabricating dental prostheses, special attention must be paid to the masticatory forces, which are the result of a complex interaction between the masticatory muscles, dental arches, maxillary structures, and specific masticatory movements [50]. The evaluation of the masticatory forces is particularly important for patients with specific conditions, such as bruxism or clenching, who develop increased mechanical stress, affecting the resistance of prosthetic restorations. The results indicate force values over 1000 N and a pressure of 40 MPa on the dental surfaces, which can generate abnormal wear levels of denture teeth [51]. The maximum bite force (MBF) depends mostly on the area of the dental arch, reaching highest values in the molar area (~800 N) and minimum levels in the incisive area (150–200 N) [52,53].

As already mentioned in the present study, the intraoral use of zirconia as a restorative material is also associated with, apart from possible fractures, complications such as chemical aging and the wear of antagonist teeth. Chemical aging is also known as low temperature degradation; water can destabilize the meta-stable tetragonal zirconia grains (transition from the T phase to the M phase at room temperature, in the absence of any mechanical stress), so that the volume increase associated with phase change can cause local stresses and microcracks near the surface [54]. This, in turn, generates internal stress, which contributes to the progression of micro-fissures in the materials, causing particle detachment with an overall negative impact on material strength [23]. In the end, the material can fracture prematurely. As far as antagonist teeth wear is concerned, studies have shown that polished monolithic zirconia prosthetic restorations produce higher enamel wear rates compared to natural tooth enamel, but lower wear rates than glazed metal-ceramic crowns [55].

Several generations of zirconia have been developed until now [14,23,56], with the purpose of obtaining ceramic biomaterials with overall enhanced properties and characteristics in terms of aesthetics, mechanical properties, chemical stability, and biocompatibility, thus making them suitable for use in indirect dental restorations. The new generation of yttria-stabilized zirconia faces, however, a big challenge: how to benefit from improved aesthetics, comparable with those provided by glass-ceramics, while maintaining specific mechanical properties [23]. In this respect, one option is the infiltration of feldspathic glass in the outer surface of zirconium dioxide (ZrO_2_), which will significantly improve the aesthetic appearance of the material, but will also produce modifications of the Young modulus, with a direct influence on the tensile strength [23]. Another option is offered by the possibility of using zirconium dioxide (ZrO_2_) powder containing nanoparticles; however, due to the complex processing, it is difficult to obtain a homogeneous powder.

## 5. Conclusions

Zirconia is nowadays considered to be one of the most used bioceramic material for single- and multi-unit fixed dental restorations or for abutments and infrastructures of implant-supported prostheses.

In the present study, three-unit zirconia fixed dental prostheses manufactured by CAD/CAM technology were subjected to bending strength testing. Four types of zirconia materials were purchased from different suppliers. Our findings highlighted the following aspects: according to SEM investigations, the tested materials have different particle sizes, and defects are visible in some of them; the EDS analysis showed the presence of all specific elements in the tested zirconia samples (Zr, Hf, and Y), and no major differences in regard to chemical composition were observed among the samples; the bending tests indicated that the highest forces at the failure values were registered for the B zirconia samples, followed by K-H and N samples, while the lowest values were obtained for the K-S specimens; and bending strength proved to be in correlation with the particle size and distribution, but also with the presence of certain defects in the zirconia material. Nevertheless, all investigated materials have exhibited higher fracture toughness values than the ones clinically accepted.

Within the limitations of our study, we were able to confirm the hypothesis that the new zirconia-based materials with progressively increased translucency offer results comparable to those of materials with constant flexural strength when subjected to a three-point bending test. Nevertheless, additional scientific studies are required to confirm the results we have obtained.

In perspective, other aspects such as phase identification and distribution within the structure, usage of different sintering temperature, or other key factors that can influence the behavior of zirconia need to be thoroughly analyzed and corroborated by finite element simulations, with the aim of gathering additional scientific information and a clearer insight on these issues.

## Figures and Tables

**Figure 1 materials-14-06887-f001:**
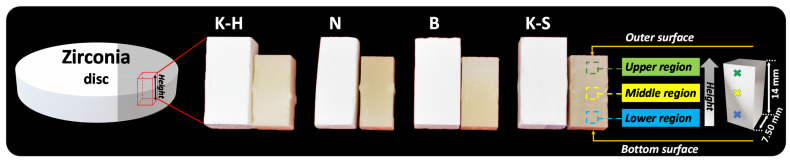
Macroscopic images of the zirconia samples before and after the sintering process.

**Figure 2 materials-14-06887-f002:**
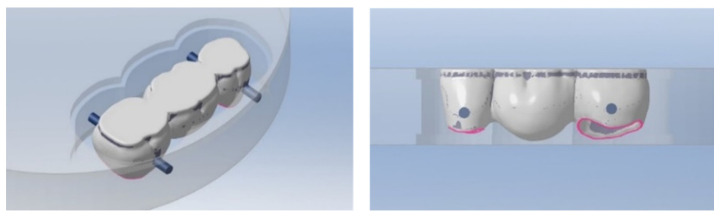
Test samples nesting process.

**Figure 3 materials-14-06887-f003:**
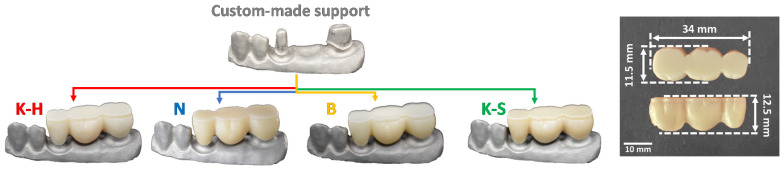
Macroscopic images of the investigated three-unit zirconia test samples.

**Figure 4 materials-14-06887-f004:**
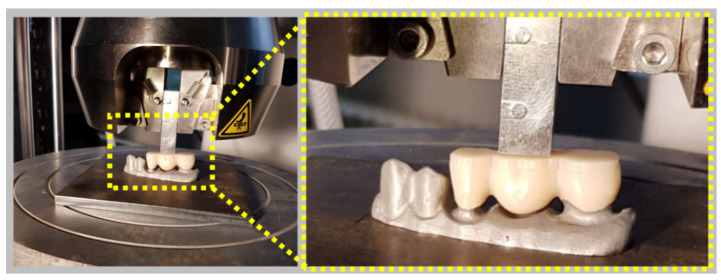
Representation of the bending test setup.

**Figure 5 materials-14-06887-f005:**
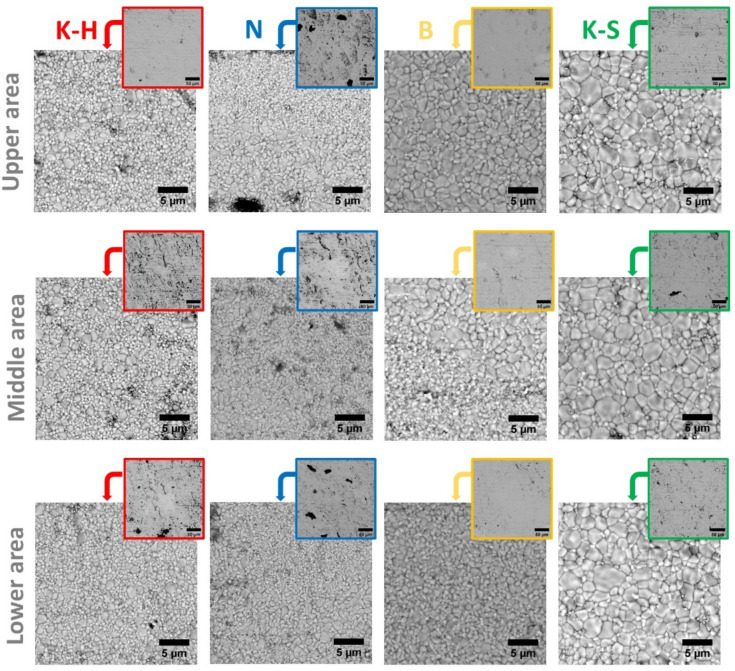
Images at different magnifications of the morphology of the zirconia specimens investigated in different regions by SEM.

**Figure 6 materials-14-06887-f006:**
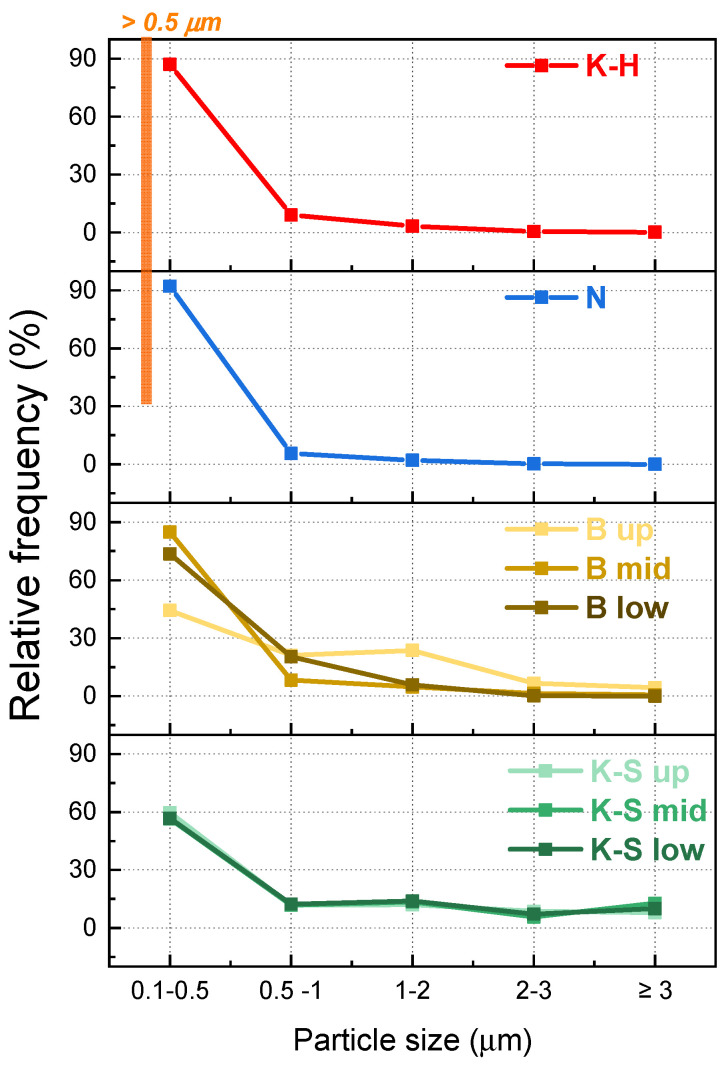
Relative frequency of the grain size distribution.

**Figure 7 materials-14-06887-f007:**
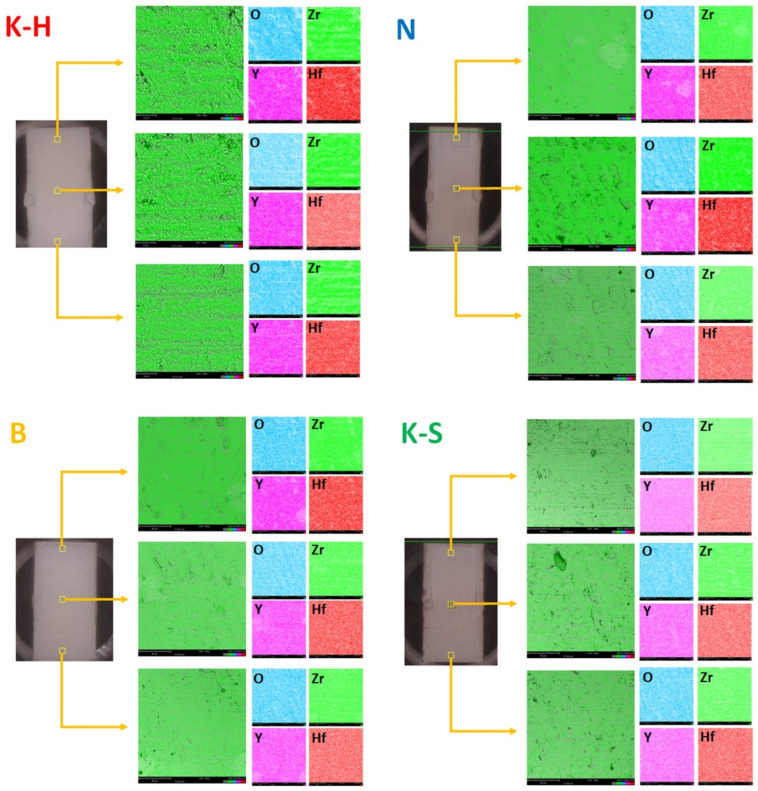
Elemental distribution of the investigated zirconia materials.

**Figure 8 materials-14-06887-f008:**
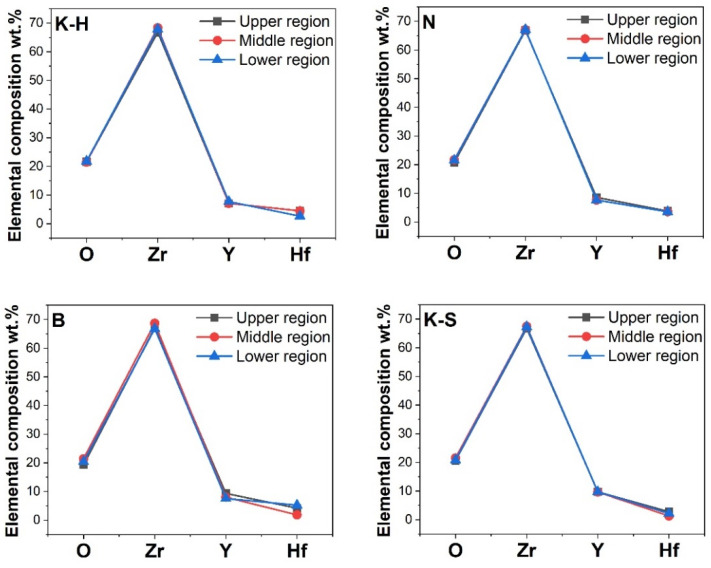
Elemental composition of the investigated materials.

**Figure 9 materials-14-06887-f009:**
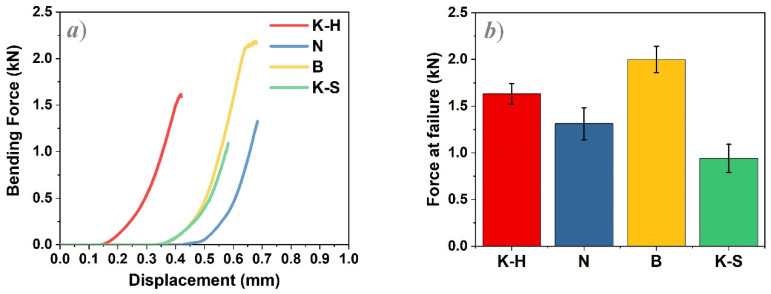
Diagrams representing the bending force–displacement curves (**a**) and the force at average failure values (**b**) for the tested zirconia three-unit dental prostheses.

**Figure 10 materials-14-06887-f010:**
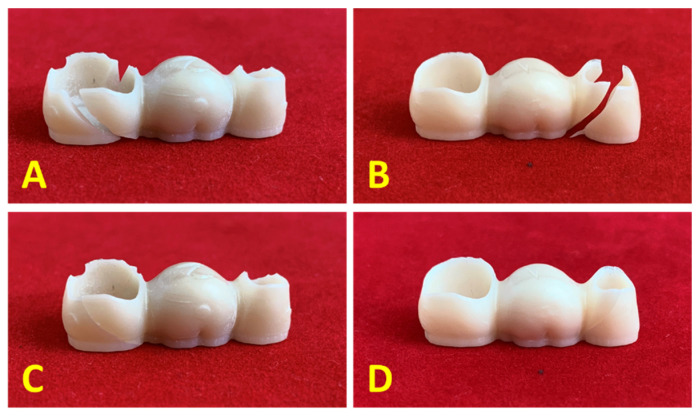
Representative macroscopic images of the fracture line of the three-unit zirconia dental prostheses (**A**,**B**—fractured pieces resulted after the bending test, **C**,**D**—fractured pieces aligned).

**Figure 11 materials-14-06887-f011:**
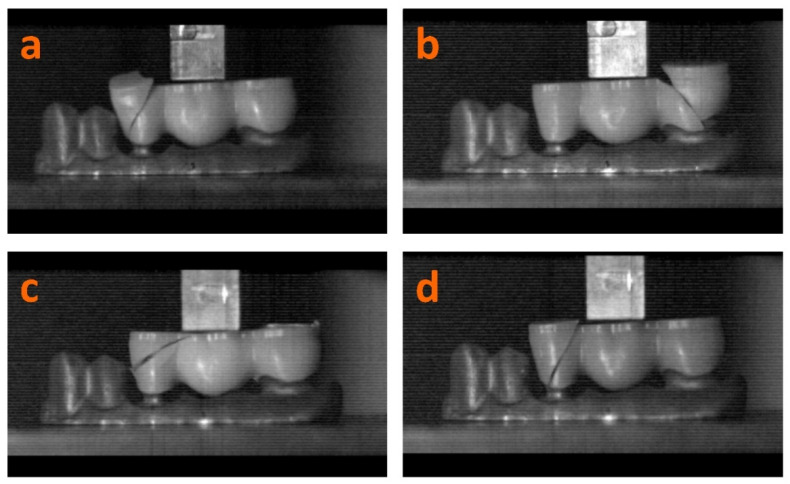
Fracture patterns for K-H (**a**), N (**b**), B (**c**), and K-S (**d**) three-unit zirconia dental prostheses.

**Table 1 materials-14-06887-t001:** Manufacturer, type, and sample codification of the tested zirconia-based materials.

Material	Manufacturer	Type	Codification
Zirconia	Kuraray Noritake Inc.	Katana^TM^ Zirconia HTML	K-H
	Novadent/Dentaltechnik	NOVAZir^®^ Fusion float^®^ ml	N
	Bloomden Bioceramics	3D PRO Zirconia	B
	Kuraray Noritake Inc.	Katana^TM^ Zirconia STML	K-S

**Table 2 materials-14-06887-t002:** Grain size distribution in the investigated materials.

Material		Relative Frequency					Average Grain Size Area (μm^2^)
		0.10–0.50	0.51–1.00	1.01–2.00	2.01–3.00	>3.01	
K-H		87.01%	9.10%	3.26%	0.49%	0.15%	0.22
N		92.10%	5.58%	2.09%	0.23%	-	0.20
B	Up	44.26%	21.11%	23.65%	6.59%	4.39%	0.89
	Mid	84.81%	8.33%	4.75%	1.40%	0.70%	0.35
	Low	73.65%	20.37%	5.82%	0.17%	-	0.37
K-S	Up	59.66%	11.82%	12.01%	8.63%	7.88%	0.95
	Mid	56.49%	11.72%	13.39%	5.65%	12.76%	1.06
	Low	56.74%	12.36%	13.86%	7.12%	9.93%	0.97

**Table 3 materials-14-06887-t003:** Comparison of the maximum force at failure values for the tested zirconia samples.

Materials	Force at Failure (kN)
K-H	1.63 (±0.11)
N	1.31 (±0.17)
B	2.00 (±0.14)
K-S	0.94 (±0.15)

## Data Availability

Not applicable.

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
