# Peer review of "Bending Fracture of Different Zirconia-Based Bioceramics for Dental Applications: A Comparative Study"

_materials, 2021, doi:10.3390/ma14226887_

Round 1

Reviewer 1 Report

There are some matters to be clarified or precisely described for readers to understand the results.

1: Grain size was expressed in the dimension of µm2  or nm or µm. To select one of the three dimensions is recommended.

2:  You claim no difference in chemical composition among the four samples on page 7, however, the total composition  of "lower side" of sample K-H shown in Figure 8 on page 9 is much less 100%. You had better explain the reason.

3: More clear explanation is required on the reason why sample B showed the highest mechanical strength with the medium average grain size, stated on page 10.

Reviewer 2 Report

The manuscript features an introduction, review of the literature, materials and methods, and results. The abstract is concise and well written. The manuscript provides the necessary background information covering 39 references. The research methodology is appropriate, but some suggestions are recommended.

1-In the “Introduction” section, it is suggested to cite and discuss more literature relevant to the topics of the present research work. Only 12/39 references are introduced.

2-About the experimental tests, how many repetitions were conducted to obviate the measuring errors, the repeatability / consistency was checked?

3-And about the equipment’s calibration?

4-Why the value of 1mm/min for the test speed?

5-The imposed force, in each test model, was applied on the ‘’pontic of the dental prostheses’’. What is your opinion about the application of this load condition? Any slipping, or friction between the part contacts?

6-About the fracture patterns (figure 10) it will interesting to compare this solution, identifying alternative tests (computational models) or verify this solution with previous research. Authors should increase their discussion with previous research.

7-In the conclusions, some future directions and research gaps need to be included.

Reviewer 3 Report

This is an interesting study and the article is well written. However, few minor mistakes need to be addressed:

  1. Grain size is mentioned (lines # 34 and 37) and at other places in the article. The unit is not correct i.e. µm2.
  2. Techniques like ‘scanning electron microscopy and energy-dispersive spectroscopy’ needs abbreviations at this point (line # 86) for further referral (e.g. at line # 105 only abbreviation should be used).
  3. Authors should include the elemental composition of the purchased types of zirconia.
  4. Authors should mention the material used for the custom made support as it is being used in three-point bend test, it is important to know the support material and its strength.
  5. Size of the zirconia dental prostheses and dimensions of parallelepiped should be included in the figures.
  6. In my opinion, the SEM images of fractured prostheses should be included in the study. Authors discuss the grain size and mechanism behind fracture. Thus, the SEM images will strengthen the argument.
  7. Reference 14 (line # 189) is not appropriately placed.
  8. The sentence “found in the middle und occlusal third of the crown (line 271)”. Here “und” is misspelled. I think authors wish to write “and”. Please revise.
  9. Revise the sentence to clarify the meaning (line # 279-282).
  10. The authors have used zirconia throughout the article. Here (line # 303, 305, and 311) zirconium dioxide and formula ZrO2 are used. In my opinion, zirconia should be used here as well to maintain the consistency. Or, the authors can use chemical formula at all places after introducing at start.
  11. Check sentence for language (line # 377-378).
  12. The article should be rechecked grammatically.
